# Dietary stearic acid regulates mitochondria in vivo in humans

Deniz Senyilmaz-Tiebe [1,2], Daniel H. Pfaff[1,3], Sam Virtue[,4], Kathrin V. Schwarz[5], Thomas Fleming[3,6,7,8], Sandro Altamura[9,10], Martina U. Muckenthaler [9,10], Jürgen G. Okun[5], Antonio Vidal-Puig[4,11], Peter Nawroth[3,6,7,8] & Aurelio A. Teleman [1,2]

Since modern foods are unnaturally enriched in single metabolites, it is important to understand which metabolites are sensed by the human body and which are not. We previously showed that the fatty acid stearic acid (C18:0) signals via a dedicated pathway to regulate mitofusin activity and thereby mitochondrial morphology and function in cell culture. Whether this pathway is poised to sense changes in dietary intake of C18:0 in humans is not known. We show here that C18:0 ingestion rapidly and robustly causes mitochondrial fusion in people within 3 h after ingestion. C18:0 intake also causes a drop in circulating long-chain acylcarnitines, suggesting increased fatty acid beta-oxidation in vivo. This work thereby identifies C18:0 as a dietary metabolite that is sensed by our bodies to control our mitochondria. This could explain part of the epidemiological differences between C16:0 and C18:0, whereby C16:0 increases cardiovascular and cancer risk whereas C18:0 decreases both.

[1] German Cancer Research Center (DKFZ), 69120 Heidelberg, Germany. [2] Heidelberg University, 69120 Heidelberg, Germany. [3] Department of Internal Medicine I and Clinical Chemistry, Heidelberg University Hospital, 69120 Heidelberg, Germany. [4] University of Cambridge Metabolic Research Laboratories, Wellcome Trust-MRC Institute of Metabolic Science, Cambridge CB2 0QQ, UK. [5] Center for Child and Adolescent Medicine, Division of Neuropediatrics and Metabolic Medicine, University Hospital Heidelberg, Im Neuenheimer Feld 669, 69120 Heidelberg, Germany. [6] German Center for Diabetes Research (DZD), 69120 Heidelberg, Germany. [7] Joint Division Molecular Metabolic Control, German Cancer Research Center (DKFZ) and Heidelberg Center for Molecular Biology (ZMBH), University Hospital Heidelberg, 69120 Heidelberg, Germany. [8] Joint Heidelberg-IDC Translational Diabetes Program, Institute for Diabetes and Cancer, IDC Helmholtz Center Munich, 69120 Heidelberg, Germany. [9] Department of Pediatric Hematology, Oncology and Immunology, University of Heidelberg, 69120 Heidelberg, Germany. [10] Molecular Medicine Partnership Unit, 69120 Heidelberg, Germany. [11] Wellcome Trust Sanger Institute, Hinxton, Cambridgeshire CB10 1SA, UK. These authors contributed equally: Deniz Senyilmaz-Tiebe, Daniel H. Pfaff. Correspondence and requests for materials should be addressed to A.A.T. (email: a.teleman@dkfz.de)

The human diet is a complex mixture of metabolites, and the composition of our diet impacts our health and the development of diseases such as diabetes, cardiovascular complications, and cancer[1,2]. Whilst many different metabolites influence human physiology and health when ingested, not all metabolites within a given class are actively sensed by the body to respond appropriately to changes in dietary intake and to maintain homeostasis. For instance, glucose elicits a strong insulin secretion response, systemically activating insulin and PI3K signaling, thereby restoring blood sugar levels to "normal", whereas fructose only elicits a weak response[3–5]. Similarly, the anabolic and oncogenic mTORC1 pathway is activated by the presence of some amino acids such as leucine, arginine, and methionine, but not by other amino acids[6]. Since it is likely unfeasible for an organism to sense all metabolites in its diet, it appears that evolution has selected certain metabolites within a class to be sensed by the organism and to act as proxies for the intake of the entire class. This sensing mechanism works in nature because natural food sources usually do not contain only single metabolites from a class, for instance leucine, but not other amino acids. Hence sensing one metabolite from a class is sufficient to indicate the presence of the entire class in the food. Modernization has changed this, however, providing humans with food sources particularly high in single metabolites such as fructose or palmitic acid. This leads to mismatches between what the body senses and what it is actually ingesting, especially when the ingested metabolite is not the one being sensed. Hence it is critical to understand which metabolites are being sensed by the human body, and what physiological responses they elici

Within the metabolite class of fatty acids, epidemiological studies have found that various fatty acids have different biological consequences when ingested. Saturated fatty acids in general, and palmitic acid (C16:0) in particular, are harmful in part because they elevate LDL cholesterol and atherosclerosis risk[7]. Dietary stearic acid (C18:0), however, does not increase atherosclerosis risk, and, if anything, actually reduces LDL cholesterol[7–10]. Indeed, increased levels of circulating C18:0 lipids are associated with reduced blood pressure, improved heart function, and reduced cancer risk[11–15]. Hence unlike other saturated fatty acids, and contrary to the general belief that saturated fatty acids are harmful, C18:0 appears to have some beneficial effects on human health. The molecular mechanisms of this, however, are not clear.

We recently reported that C18:0 regulates mitochondrial morphology and function in Drosophila and in human cells via a dedicated signaling pathway[16]. When C18:0 levels are low, the Transferrin Receptor TfR1 activates JNK signaling, leading to ubiquitination and inhibition of Mitofusin 2 and hence mitochondrial fragmentation and reduced oxygen consumption[16]. In the presence of C18:0, the fatty acid molecule is covalently attached to TfR1 via a thioester bond in a post-translational modification called stearoylation, analogous to protein palmitoylation by C16:0. This leads to reduced JNK activation by TfR1, to mitochondrial fusion, and to elevated oxygen consumption[16]. Hence, in cell culture, cells sense the levels of C18:0 and respond via a signaling pathway that ends in mitochondrial activation. Unexpectedly, feeding C18:0 to Drosophila animals also regulated mitochondrial fusion in vivo, and led to organismal consequences: dietary supplementation with C18:0 was able to partially rescue the impaired locomotion and reduced lifespan resulting from mitochondrial dysfunction caused by mutations in the Parkinsons genes Pink1 or parkin[16]. This indicates that flies can sense and respond to levels of C18:0 in their diets. Humans, however, ingest more lipids than flies[17,18]. Furthermore, humans, like flies, express Elovl6, which elongates C16:0 to C18:0 thereby providing an endogenous source of C18:0. Hence it is unknown whether in humans this C18:0 sensing mechanism is saturated

with dietary and/or endogenous lipids, or if it is poised to sense changes in dietary C18:0 intake, responding by altering mitochondrial morphology and function. To test this, we undertook here a clinical study, manipulating dietary C18:0 intake in both healthy and diabetic subjects, and assayed mitochondrial morphology and function in vivo. We found a surprisingly robust response, with C18:0 ingestion causing fusion of mitochondria in 90% of all tested subjects. C18:0 intake also led to a drop in circulating acylcarnitine levels, suggesting increased fatty acid beta-oxidation in vivo. This identifies C18:0 as a lipid metabolite sensed in humans, and raises the possibility that dietary C18:0 activates part of the physiological response through which the human body handles lipids.

## Results

**C18:0 ingestion causes mitochondrial fusion.** We aimed to study whether acute changes in C18:0 intake in humans affect mitochondrial morphology and function in vivo. C18:0 is enriched in meat, in dairy products, and in certain seeds such as cocoa beans. Hence we first asked subjects to eat for 2 days a low-fat vegan diet which is low in C18:0, to bring everyone to a low-C18:0 baseline regardless of their habitual diet (Fig. 1a). We then asked the subjects to drink a banana milk shake containing 24 g of C18:0, roughly equivalent to the amount of C18:0 present in 200 g of milk chocolate[19]. To assay mitochondrial morphology, we chose to study neutrophils because, unlike other tissues, blood can be obtained in a minimally-invasive manner, and neutrophils are the most abundant blood cell type that has mitochondria. We took one "basal" blood sample prior to the 2-day low-C18:0 diet, one blood sample directly prior to the C18:0 drink ("0 h"), and then two blood samples at 3 and 6 h after C18:0 ingestion (Fig. 1a). The 3 and 6 h timepoints were selected because a previous study found [13]C-labeled C18:0 levels peak in blood 3 h after ingestion[20]. For every blood sample, we scored neutrophil mitochondrial morphology by staining whole blood immediately after collection with CD15 and CD16 to mark neutrophils, and mitotracker to mark mitochondria (Fig. 1b). We then classified neutrophils as having either "fragmented", "intermediate", or "fused" mitochondria as shown in Fig. 1b. Our study was designed as a cross-over study, with each subject repeating this procedure twice, once with a C18:0 banana milk shake, and once with a mock banana milk shake lacking the C18:0, in randomized order (Fig. 1a). The entire study was double-blinded, so all data collection and analysis was done without the knowledge of sample identity. Both the basal and the 0 h blood samples were collected in the morning without breakfast to obtain standardized clinical blood parameters. We recruited 10 healthy volunteers and 11 type-2 diabetic patients (Supplementary Fig. 1), to test if diabetic patients respond differently, given that diabetic patients have altered lipid metabolism and mitochondrial function compared to healthy subjects[21,22]. The baseline characteristics for all subjects are presented in Supplementary Data 1–2.

After 2 days of a low-C18:0 diet, the mitochondria in neutrophils are quite fragmented; 50% of all neutrophils had fragmented mitochondria and fewer than 10% had fused mitochondria prior to ingesting the C18:0 drink ("0 h", Fig. 1c). This was true in all 21 subjects combined (Fig. 1c), in the healthy subjects (Fig. 1e) and in the type-2 diabetic patients (Fig. 1g). Ingestion of the C18:0 drink caused mitochondria to fuse, both 3 and 6 h after ingestion: the fraction of neutrophils with fragmented mitochondria dropped significantly from 50 to 25% while the fraction of neutrophils with fused mitochondria increased significantly from 7 to 27% (Fig. 1c). This occurred in both healthy and diabetic subjects (Fig. 1e, g), and the response was quite robust, with 19 of the 21 subjects (90% of subjects)

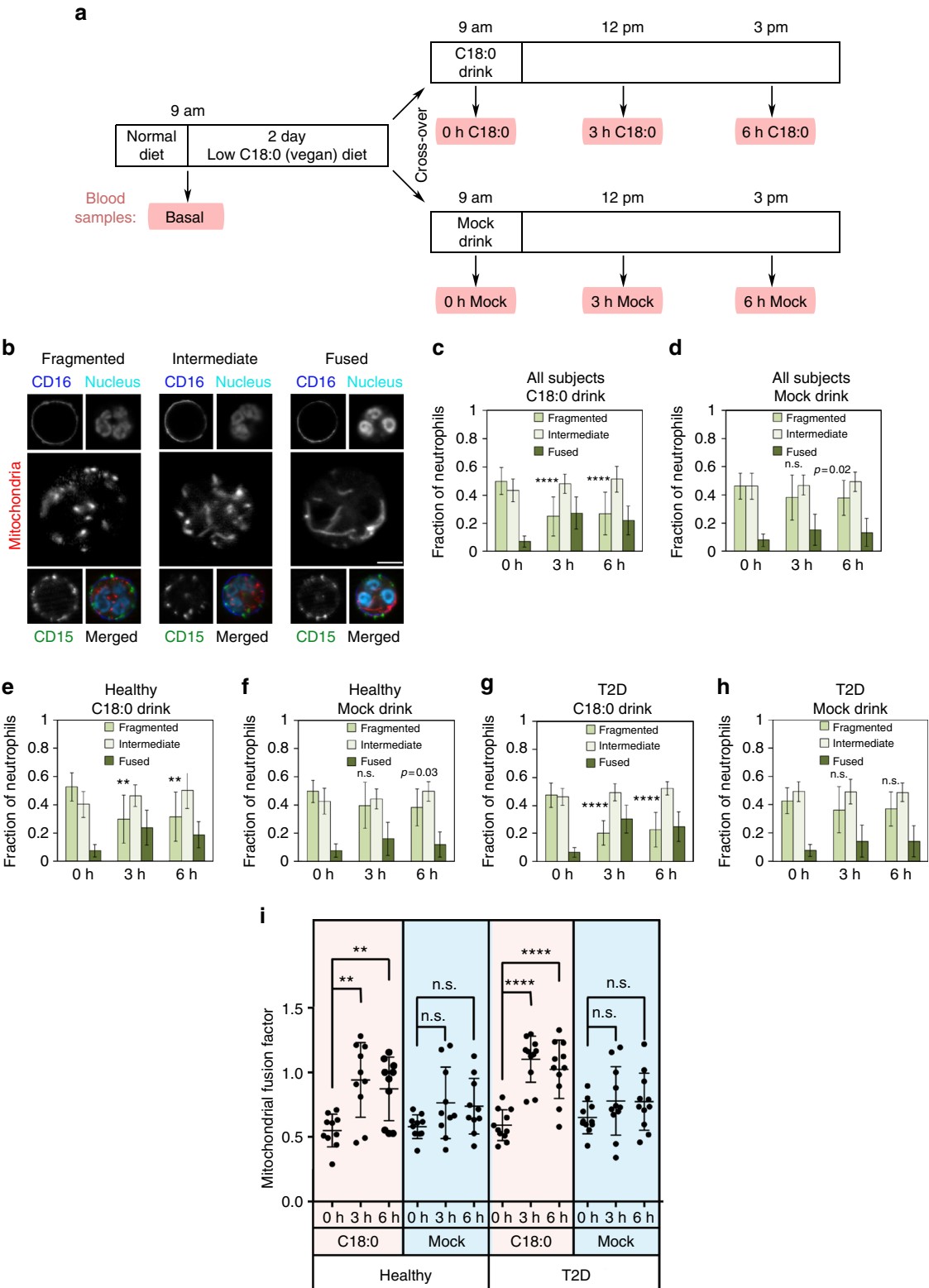

having more fused mitochondria after ingestion of C18:0. In contrast, the mock drink lacking C18:0 did not cause significant mitochondrial fusion in the very same people (Fig. 1d, f, h and Supplementary Fig. 2 for a statistical analysis comparing C18:0 to mock, using simplex plots on the entire distribution). Although not significant, the mock drink did cause a small but consistent increase in mitochondrial fusion (Fig. 1d), perhaps because the

banana + milk used for the shake contain roughly 0.5 g of C18:0. To display data for each individual subject, and to perform subsequent regression analyses, a "mitochondrial fusion factor" was calculated for each sample (Fig. 1i, for the calculation see figure legend and Methods). Analysis of this factor confirmed that mitochondria became more fused upon ingestion of the C18:0 drink, but not the mock drink, in both healthy and T2D groups

**Fig. 1** Dietary C18:0 induces mitochondrial fusion in human neutrophils. **a** Scheme of the clinical cross-over study. Subjects followed a low-C18:0 (vegan) diet for 2 days and then were given a banana milk shake either containing or lacking C18:0 (24 g) in a randomized fashion. Blood samples were collected at indicated time points. Each subject underwent the whole procedure twice, once with each milk shake. Subjects received a standardized low-fat lunch after the 3 h timepoint. The study was double-blinded. **b** Classification of neutrophils into three classes depending on mitochondrial morphology—fragmented, intermediate, or fused. Whole blood was stained with CD15 (green) and CD16 (blue) to mark neutrophils, DRAQ5 (cyan) to stain nuclei and mitotracker (red) to score mitochondrial morphology. Scale bar = 25 μm. **c–i** Ingestion of C18:0 causes mitochondria of neutrophils to fuse, seen in **c–h** as a decrease in the fraction of neutrophils with fragmented mitochondria and an increase in fused mitochondria or in **i** as an increase in the mitochondrial fusion factor. **c**, **d** All 21 subjects. **e**, **f** 10 healthy volunteers. **g**, **h** 11 type-2 diabetic patients. **i** Same data as in **c–h**, but shown by calculating a "mitochondrial fusion factor" (2 × fraction of fused mitochondria + 1 × fraction of intermediate mitochondria). Each dot represents one subject. (Error bars = std. dev., $n = 55$ neutrophils per subject per timepoint. **$p \leq 0.01$, ****$p \leq 0.0001$ by $t$-test.)

(Fig. 1i). Together, these data indicate that dietary C18:0 rapidly induces mitochondrial fusion in human neutrophils. Although we did not assay mitochondrial morphology in other tissues, we previously published that various different human and *Drosophila* cell types respond to C18:0 in tissue culture[16]. Hence we believe it is possible that other cells in the human body may also fuse their mitochondria after a meal containing C18:0, although future work will be required to test this.

**Blood C18:0 levels correlate with mitochondrial morphology.** Since neutrophil mitochondria fuse after ingestion of C18:0, this suggests serum levels of C18:0 increased after C18:0 feeding. To this end, we quantified the fatty acids present in serum triacylglycerides (TAG, Fig. 2a–d and Supplementary Fig. 3). As expected, the level of C18:0 in serum triglycerides (C18:0-TAG) dropped significantly after 2 days of low-fat vegan diet (Fig. 2a), and increased upon ingestion of the C18:0 drink, but not the mock drink (Fig. 2b, c). Interestingly, C16:0-TAG levels did not drop after 2 days of the diet, despite the diet being low-fat (Fig. 2d), suggesting that levels of circulating C16:0 are more buffered in our bodies. Circulating C16:0-TAG also did not increase in response to C18:0 feeding ("3 h" vs "0 h", Fig. 2d). Together, these data indicate that circulating levels of C18:0 are responsive to acute changes in diet, and that they are more responsive than C16:0.

In agreement with our finding that C18:0 induces mitochondrial fusion in cell culture[16], serum C18:0-TAG levels correlated significantly with the mitochondrial fusion factor across all subjects, diets, and timepoints ($p < 0.0001$, Fig. 2e–h). This correlation was specific to C18:0, as there was no significant correlation between C16:0-TAG and mitochondrial fusion factor (Fig. 2i). We wanted to test if there are any additional factors affecting mitochondrial morphology besides C18:0. Hence in addition to fatty acids, we also measured other metabolites and markers in the blood samples including cholesterol, glucose, insulin, methylglyoxal (MG), iron, ferritin, transferrin, and hepcidin. None of them showed C18:0-dependent changes in levels (Supplementary Figs. 4–5). Interestingly, the low-fat vegan diet increased hepcidin levels (Supplementary Fig. 5), but additional work will be needed to understand the significance of this. A multivariate regression analysis identified only C18:0-TAG as significantly contributing to determine the mitochondrial fusion factor (Supplementary Data 3). We noticed that the mitochondria in some people fused particularly strongly in response to C18:0 ingestion, whereas the mitochondria in 10% of the people did not respond. Hence we correlated the mitochondrial response upon C18:0 ingestion (i.e., change in mitochondrial fusion factor 3 h vs 0 h) to 44 different clinical parameters, but there were no significant correlations with any of the measured parameters after accounting for multiple testing (Supplementary Data 4).

To make sure that the mitochondrial fusion elicited by the C18:0 banana milk shake was due solely to the C18:0, and not to a

combined action of C18:0 with other components of the milk shake, we asked one subject in a pilot study to ingest pure C18:0 emulsified in drinking water and scored the subject's mitochondrial morphology. As expected, ingestion of pure C18:0 significantly induced mitochondrial fusion in vivo (Supplementary Fig. 6a).

Two lines of reasoning suggested this mitochondrial fusion response should be specific for C18:0. Firstly, epidemiological studies have shown that palmitic acid (C16:0) and stearic acid (C18:0) have very different effects on physiology and health[7,11–15]. Secondly, our previous work in cell culture showed that mitochondrial fusion is very specifically induced by C18:0 and not by any other fatty acid such as C16:0, C18:1, or C20:0, because there is a dedicated signaling pathway that senses C18:0[16]. To confirm the specificity of this response in vivo, we first asked the same subject to ingest a banana milk shake containing C16:0 (equimolar amount as in the C18:0 drink). Unlike the C18:0 drink, the C16:0 drink did not induce mitochondrial fusion (Supplementary Fig. 6b). C16:0 ingestion also did not increase C16:0-TAG levels in blood (Supplementary Fig. 6c). This suggests that serum C16:0-TAG levels are more buffered than C18:0-TAG levels, in agreement with the fact that C16:0-TAG levels also do not drop after 2 days of a low-fat diet (Fig. 2d). To increase the statistical power of our C16:0 findings, we re-recruited six healthy subjects from our main study, and asked them to follow a low-fat vegan diet for 2 days. In a double-blinded fashion, four of these subjects then received a banana milk shake containing C16:0 (equimolar to the C18:0 drink). As a positive control, one subject received a C18:0 drink, and as a negative control one subject received a mock drink. We then scored neutrophil mitochondrial morphology at 3 h post-ingestion, since this timepoint showed the largest effect in the main study. This confirmed that all five subjects receiving a C16:0 drink did not fuse their mitochondria (Supplementary Fig. 6f), and the positive and negative-control subjects responded as anticipated (Supplementary Figs. 6g–h). Since 90% of all subjects in the main study responded to C18:0 by fusing their mitochondria, this means the lack of mitochondrial fusion in response to C16:0 in all six tested subjects (Supplementary Figs. 6b and 6f) is highly significant ($p = 10^{-5}$). Thus the mitochondrial fusion caused by C18:0 is specific to C18:0 and not caused by C16:0 ingestion.

**C18:0 ingestion reduces circulating acyl-carnitines.** We next asked if there is a functional consequence of C18:0 ingestion. We previously reported that reduced C18:0 levels lead to reduced mitochondrial respiration in *Drosophila*[16], possibly due to reduced mitochondrial fatty acid beta-oxidation. To test if beta-oxidation is affected, we measured long chain acylcarnitine levels in the blood samples. Acylcarnitines are fatty acids coupled to carnitine that are ready for mitochondrial import for beta-oxidation. When beta-oxidation is impaired relative to the fatty acid supply, these long chain acylcarnitines accumulate and are released into the blood where they can be detected, forming the

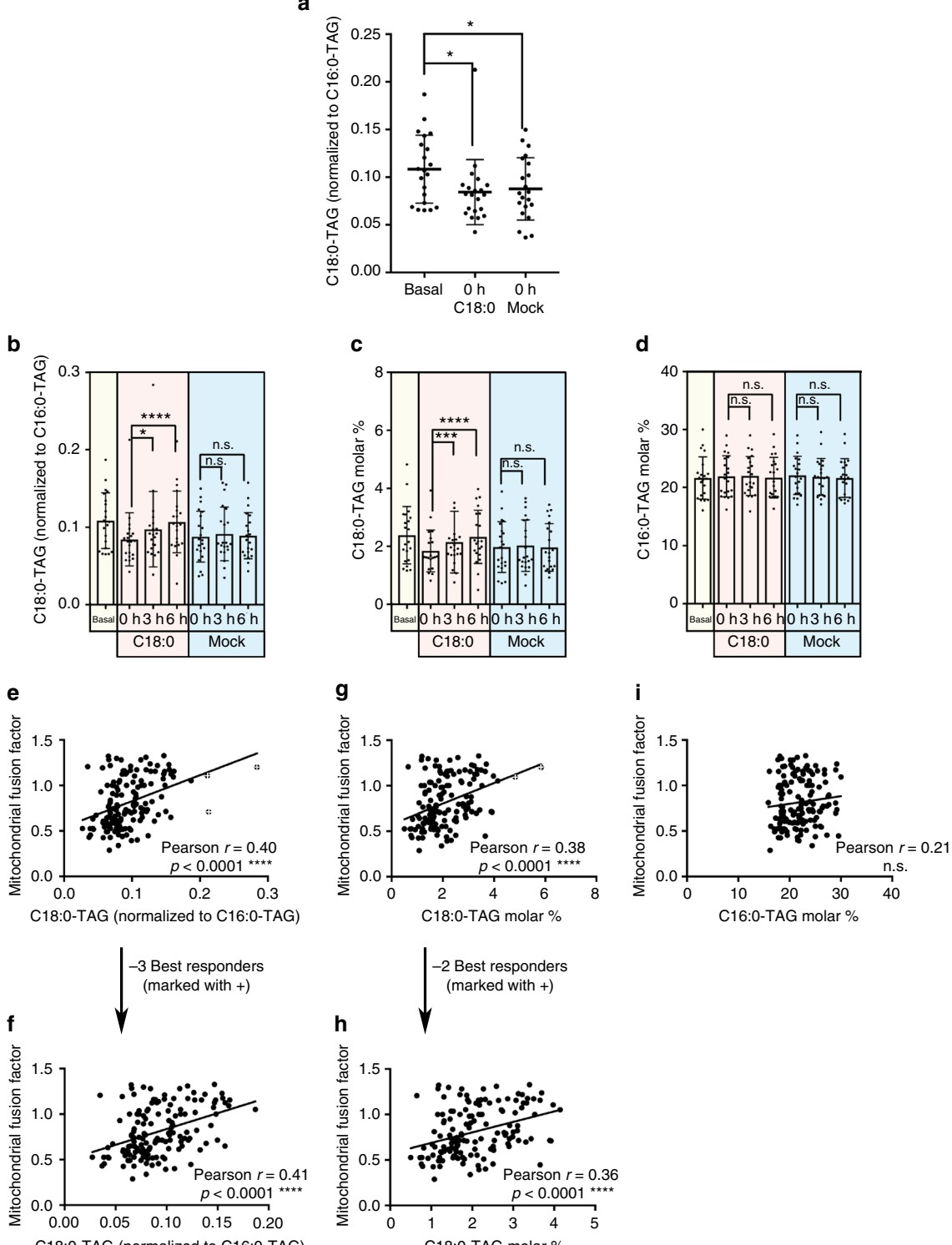

**Fig. 2** Serum C18:0-TAG levels are responsive to dietary C18:0 intake and correlate with mitochondrial fusion. **a** Levels of serum C18:0 present in triglycerides ("C18:0-TAG") significantly drops after 2 days of low C18:0 diet. (Data normalized to C16:0-TAG levels, $n = 21$ subjects, $*p \leq 0.05$ by $t$-test.) **b**, **c** Serum C18:0-TAG levels increase after consuming the C18:0 drink but not the mock drink. **b** C18:0-TAG normalized to C16:0-TAG, since C16:0 is abundant and circulating levels are stable. **c** Molar percentage C18:0-TAG. ($n = 21$ subjects, error bars = std. dev. $*p \leq 0.05$, $***p \leq 0.001$, $****p \leq 0.0001$ by $t$-test) **d** C18:0 intake does not affect C16:0-TAG levels. ($n = 21$ subjects, error bars = std. dev.) **e–i** Mitochondrial fusion in neutrophils correlates with serum C18:0-TAG levels (**e–h**) but not C16:0-TAG levels (**i**) across all subjects, timepoints, and dietary conditions. ($n = 21$ subjects × 7 timepoints = 147 serum samples)

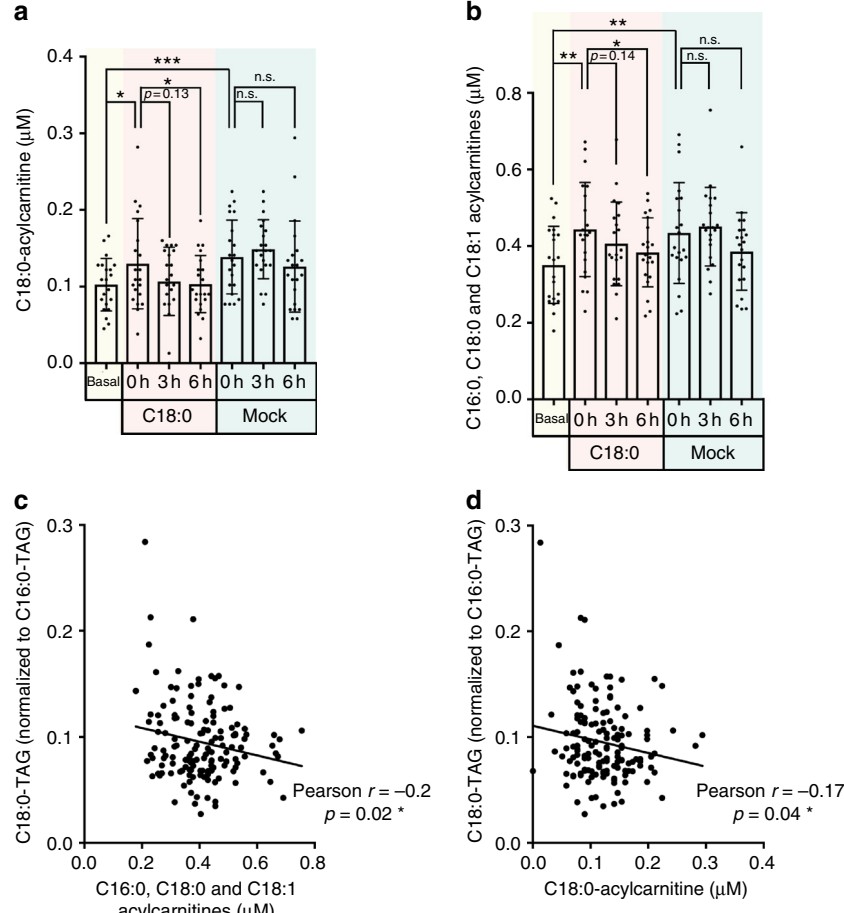

**Fig. 3** Dietary C18:0 intake activates fatty acid beta-oxidation in vivo. **a**, **b** Serum levels of **a** C18:0-acylcarnitine or **b** long-chain-fatty-acid-acylcarnitines, markers for reduced fatty-acid beta-oxidation, drop upon dietary intake of C18:0. Note that the non-significant drop at 6 h in both C18:0 and mock conditions can be due to eating lunch after the 3 h timepoint. ($n$ = 21 subjects, error bars = std. dev. *$p \leq 0.05$, ***$p < 0.001$ by $t$-test.) **c**, **d** Serum levels of **c** C18:0-acylcarnitine or **d** long-chain-fatty-acid-acylcarnitines, markers for reduced fatty-acid beta-oxidation, anti-correlate with serum C18:0-TAG levels across all subjects, timepoints, and dietary conditions. ($n$ = 21 subjects × 7 timepoints = 147 serum samples)

basis for screenings of newborn babies for beta-oxidation defects[23]. Interestingly, we found that serum long chain acylcarnitine levels increase significantly after a 2-day low-fat vegan diet ("0 h" vs "basal", Fig. 3a, b). Conversely, long chain acylcarnitine levels dropped significantly after ingesting the C18:0 drink but not the mock drink (Fig. 3a, b), suggesting that C18:0 intake increases fatty acid beta-oxidation in vivo. Particularly striking, C18:0 ingestion leads to an increase in circulating C18:0-TAG levels (Fig. 2), but causes a decrease in circulating C18:0 acylcarnitine levels (Fig. 3a), indicating that the acylcarnitine drop is not due to reduced substrate availability but rather to increased acylcarnitine usage. In the mock treatment, there was no drop in acylcarnitine levels at 3 h, however at 6 h there was a non-significant trend towards reduced levels. This could be due to the fact that all subjects received a low-fat vegan lunch after the 3 h timepoint to avoid hunger-induced artefacts. Furthermore, we also found a significant, negative correlation between long chain acylcarnitine levels and C18:0-TAG levels across all subjects, diets, and timepoints (Fig. 3c, d). In contrast, C16:0 ingestion did not lead to a drop in acylcarnitine levels (Supplementary Fig. 6d–e), indicating that the effect of C18:0 on acylcarnitines is specific. Together, these data indicate that C18:0 ingestion leads to a physiological response in vivo.

**C18:0 effects in the medium-term**. The data thus far relate to the main goal of this study, which was to determine whether an acute

increase in dietary C18:0 intake can induce mitochondrial fusion in vivo in humans. This appears to be the case, indicating that this signaling pathway in humans is indeed tuned to sense changes in C18:0 intake. This raises the question whether people with different dietary habits, and hence different long term C18:0 intake levels, have long-term differences in their mitochondrial morphology and function. To this end, we compared our "basal" data, prior to the 2-day low-C18:0 diet, to the "0 h" data on both the days the subjects received the C18:0 drink or the mock drink (Fig. 4a). Interestingly, mitochondria were consistently more fragmented (lower mitochondrial fusion factor) after 2 days of low-C18:0 diet in all subject groups—healthy volunteers, type-2 diabetics, vegetarians, and omnivores (Fig. 4a). This is consistent with the reduced C18:0-TAG levels in serum after 2 days of low-C18:0 diet (Fig. 2a) and suggests that dietary C18:0 intake can affect circulating C18:0 levels and mitochondrial morphology not only acutely but also within a medium-term 2-day timeframe. This result is particularly striking given that both the "basal" blood samples and the "0 h" blood samples were collected after overnight fasting, indicating that the C18:0 effect persists for hours. We had four vegetarians in our study (Supplementary Data 1). Since vegetarians eat no meat, but still eat dairy products, both of which are rich in C18:0, there is the possibility that vegetarians could have reduced serum C18:0 levels. However, we did not see any differences between vegetarians and omnivores in either basal C18:0-TAG levels (Fig. 4c, d) or mitochondrial fusion (Fig. 4b). This lack of an association could be due

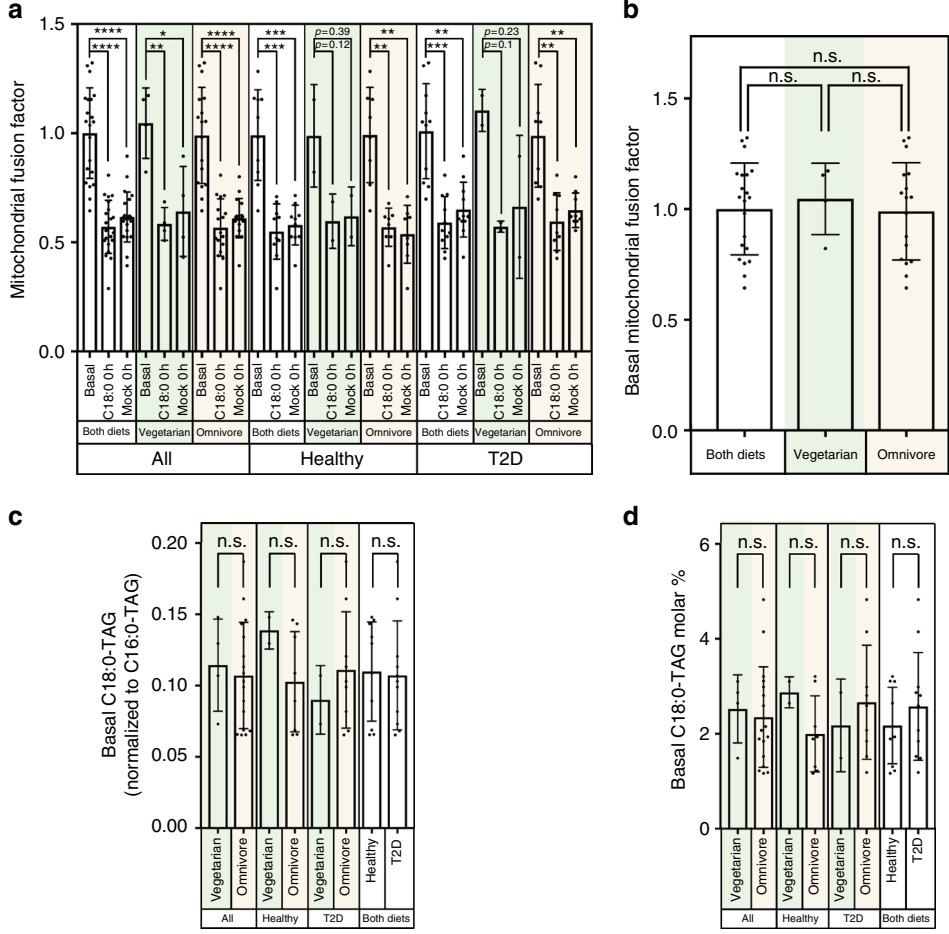

**Fig. 4** Mitochondrial morphology is affected by short and medium-term, but not long-term, differences in dietary C18:0 intake. **a** Mitochondrial fusion drops after a 2-day low-C18:0 vegan diet in all categories of subjects (vegetarian, non-vegetarian, diabetic, healthy). **b** Basal mitochondrial fusion is similar in vegetarians and omnivores. ($n = 21$ subjects total, of which eight healthy omnivores, two healthy vegetarians, nine type-2 diabetic omnivores, two type-2 diabetic vegetarians. Error bars = std. dev. *$p \leq 0.05$, **$p < 0.01$, ***$p < 0.001$, ****$p < 0.0001$ by $t$-test.) **c**, **d** "Basal" serum C18:0-TAG levels (at the beginning of the study, prior to the 2 day vegan diet) are not lower in vegetarians compared to omnivores. Note that all blood is drawn at 9:00 am after overnight fasting, which could mask dietary differences between vegetarians and omnivores. ($n = 21$ subjects total, of which eight healthy omnivores, two healthy vegetarians, nine type-2 diabetic omnivores, two type-2 diabetic vegetarians. Error bars = std. dev. n.s. = not significant by $t$-test)

to the small sample size, however, it could also be due to compensatory intake of C18:0 from dairy products in vegetarians, or from increased endogenous biosynthesis. Hence it is possible that long-term constitutive differences in dietary C18:0 intake might be buffered by changes in endogenous biosynthesis. Important to note, however, is that all our blood samples were taken after overnight fasting and no breakfast in the morning, hence they represent "baseline" levels of circulating C18:0 in the absence of recent food ingestion. The data in Fig. 2 show that circulating C18:0 levels increase after eating C18:0. Hence if people were sampled at random times of the day, there would likely be a difference in vegetarians vs omnivores due to the different C18:0 content of the food they consume. Future work will be required to address this question.

## Discussion

In this study, we identify stearic acid (C18:0) as a metabolite that is sensed in our diets and regulates human physiology, in particular mitochondrial morphology and function. Intriguingly, our data imply that when we eat, the C18:0 in our food causes our mitochondria to fuse within a few hours of eating. This response is impressively robust: we obtained statistically significant results

with only 10 healthy subjects. Unlike C18:0, C16:0 does not have this effect. This could explain part of the difference between C16:0 and C18:0 observed epidemiologically, whereby C16:0 increases the risk for cancer and cardiovascular risk whereas C18:0 reduces both[7,11–15]: if dietary C18:0 signals the intake of lipids to the human body, to activate a physiological response for lipid handling which includes fatty acid beta-oxidation, whereas C16:0 does not, this would imply that C16:0 ingestion will lead to more fat accumulation in the body than C18:0 ingestion. Fat accumulation, in turn, is a risk factor for both cardiovascular disease and cancer. Hence the balance of C16:0 to C18:0 in our diets may be important. A diet rich in C16:0 could be particularly bad because it provides lipids to the body without activating the mitochondrial response that C18:0 does. We included both healthy subjects and type-2 diabetic patients in our study, however we did not see significant differences in the basal mitochondrial morphology (Fig. 4a) or in the response to C18:0 ingestion in these two groups (Fig. 1). If anything, C18:0 ingestion caused more robust mitochondrial fusion in diabetic patients than in controls. Further work will be necessary to test whether longer-term dietary interventions with C18:0 can affect baseline levels of C18:0 in the serum, and hence mitochondrial morphology and function.

## Methods

**Design of the cross-over clinical study**. The study was conducted at the Department of Endocrinology of Heidelberg University Hospital, and was approved by the local ethics committee (Ethics Committee of the Medical Faculty of Heidelberg: Trial Code Number S-675/2015). The study was registered at ClinicalTrials.gov with identifier NCT02957838. Study participants provided written informed consent.

The study size was designed in consultation with a biometrician, Johannes Krisam (in the Acknowledgements), based on data from a pilot study, and using formulas for cross-over studies from ref. [24]. These calculations showed that for reaching a significance of 0.01 and a power of 95%, 10 participants per group are required. We therefore screened 284 outpatients of the Department of Endocrinology (see Supplementary Fig. 1), 108 of which were eligible and thus were contacted. Assuming there would be roughly a 50% drop-out rate, we recruited 20 subjects per group who gave oral consent. In the end 12 healthy controls and 11 type-2-diabetics gave written consent. The recruited type-2 diabetic patients did not require insulin treatment and did not have severe complications yet. Neither the type-2-diabetics nor the control participants suffered from other severe diseases such as cancer or anemia. The baseline characteristics of the groups are provided in Supplementary Data 1–2. Two healthy controls preferred not to repeat the second arm of the cross-over study, but did not withdraw consent concerning data generated in the first round. Hence data from the remaining 10 healthy controls that completed both arms of the study were analyzed.

Basal blood samples were collected at 9 a.m. after subjects had been eating a regular diet, followed by an overnight starvation and no breakfast. The subjects were then asked to follow a low-fat, vegan diet (low C18:0 diet) for 2 days and to maintain a diary of the food and drink they consumed, thereby allowing participants to self-report violations of the diet rules. On the day of the first timecourse, "0 h" blood samples were collected at 9 a.m. after an overnight fast and no breakfast. Immediately afterwards the subjects received a banana milk shake with or without 24 g C18:0 (see below for the recipe). At 12 p.m., the "3 h" blood samples were collected and the subjects then received a standardized low-fat vegan lunch. At 3 p.m., "6 h" blood samples were collected. Each subject underwent the whole procedure twice for the cross-over timecourse, once with the C18:0 milk shake and once with the mock milk shake lacking C18:0, in a randomized order and with a wash-out phase of at least 3 days between experiments. The timecourse intervals were determined based on ref. [20]. The timecourse was performed with a maximum of three subjects per day because the blood samples had to be processed and imaged fresh and soon after collection. The blood samples were labeled with codes and only the person who drew the blood samples knew the identity of the samples. The whole study was performed in a double-blind fashion—i.e., the people who processed the samples did not have any information about the samples. After all data were collected and all analyses were performed, the data were unblinded.

**Preparation of the banana shakes**. For the C18:0 drink, 24 g of C18:0 intended for human consumption (stearic acid, Sigma-Aldrich W303518) were mixed with 100 ml of low-fat milk. This amount of C18:0 was selected by calculating the C18:0 content of a high fat meal[19]. The mixture was microwaved to emulsify the fatty acid. One small banana was added and the whole mixture was homogenized with a hand blender. The volume was completed to 250 ml with more low-fat milk. The mock drink was prepared the same way without fatty acids. For the C16:0 drink, 21.56 g (an equal molar amount to 24 g of C18:0) of C16:0 intended for human consumption (palmitic acid, Sigma-Aldrich W283207) was added instead. Milk-shakes were prepared approximately 90 min before consumption.

**Sample collection**. Blood was drawn with butterfly needle sets at each timepoint from a different location. Several tubes were filled with different blood amounts for the following purposes: (1) one was filled with 4.5 ml to extract a leukocyte pellet after erythrocyte lysis, which was stored at −80 °C, (2) a smaller EDTA tube (Sarstedt S-Monovette® 2.7 ml K3E) was filled with 1 ml of blood for mitochondrial morphology scoring via fluorescence microscopy, (3) to keep serum samples, 4 ml of blood were drawn into a Sarstedt S-Monovette® 7.5 ml Z tube, then centrifuged at 2500$g$ for 10 min, aliquoted and stored at −80 °C, (4) for analysis of insulin, 2 ml of blood were drawn into a Sarstedt S-Monovette® 7.5 ml Z-Gel tube, and (5) for other serum parameters 2 ml of blood were put into a Sarstedt S-Monovette® 7.5 ml LH-Gel. Both tubes #4 and #5 were submitted to the high-throughput analysis facilities of the Heidelberg University Central Laboratory, which is accredited according to DIN EN ISO 15189. On the "basal" day, an additional Sarstedt S-Monovette® 2.7 ml K3E tube was filled with 2 ml for high throughput cell counting via a Siemens ADVIA 120 Hematology system and measurement of HbA1c by a Biorad Variant II Turbo.

**Staining and imaging of neutrophil mitochondrial morphology**. Blood was collected into EDTA containing tubes (Sarstedt S-Monovette® 2.7 ml K3E). An aliquot of 100 μL whole blood was put into a 1.7 ml tube and the following antibodies and dyes were added: CD15-FITC (BD Pharmingen 555401, Lot: 5267530, 1:10 dilution), CD16-BV421 (BD Pharmingen 562878, Lot: 6007759, 1:20 dilution), Mitotracker Red (Cell Signaling #9082, 200 nM final) and DRAQ5 (Abcam

ab108410, 10 μM final). CD15 and CD16 are neutrophil cell surface markers and DRAQ5 is a nuclear stain. Mitotracker red was used to image mitochondria. Staining was performed at room temperature for 15 min. Afterwards, 10 μL stained blood was put on a glass slide and covered with a coverslip. The resulting smear was immediately visualized with a fluorescence microscope. Importantly, this method allowed us to score mitochondrial morphology within 30 min of blood collection. Mitochondrial morphology of approximately 50 CD15, CD16+ neutrophils per subject per timepoint were scored. Cells containing a single large interconnected network were scored as "fused", cells containing individual dots were score as "fragmented", and all other cells were scored as "intermediate". To ensure consistency of scoring between samples, all blood samples in this paper were scored in a blinded fashion by one person. Four samples were also scored independently by a second person, arriving at qualitatively similar results (Supplementary Fig. 7).

**Calculation of mitochondrial fusion factor**. The fraction of neutrophils with fragmented, intermediate, and fused mitochondria were calculated and then the following formula was used: mitochondrial fusion factor = 1 × fraction of neutrophils with intermediate mitochondria + 2 × fraction of neutrophils with fused mitochondria.

**Statistical analyses**. Data sampling: Data from the two healthy subjects who did not complete both arms of the study were not used. Otherwise, for all statistical analyses, all available data were used and none were excluded.

Student $t$-tests were used to test significance in Figs. 1c–i, 2a–d, 3a, b, 4a–d, Supplementary Fig. 3, Supplementary Fig. 4, Supplementary Fig. 5, and Supplementary Fig. 6f. For all these figures except Fig. 4b–d, this was done using a 2-tailed, paired $t$-test because the comparisons are between two timepoints for the same subject (i.e., paired). For Fig. 4b–d, a 2-tailed unpaired student's $t$-test was used because the comparison is between subjects. For Fig. 1c–h, and Supplementary Fig. 6f, 55–60 neutrophils were scored for mitochondrial morphology per blood sample. These data were used to calculate the percentage of neutrophils in that sample with fragmented, intermediate or fused morphology. These percentages from the various samples were combined to generate the plots shown in figures. For instance, in Fig. 1c at 0 h, the "fragmented" bar represents the average of 21 datapoints, corresponding to the percentage of neutrophils with fragmented mitochondria from the 21 subjects, and the error bars represent the standard deviation of these 21 datapoints. Significance was tested by applying the student $t$-test to the "fragmented" category, comparing the 3 h or the 6 h timepoints to the 0 h timepoint (i.e., 21 datapoints for the 3 h or the 6 h timepoint, compared to the 21 datapoints for the 0 h timepoint.) For Figs. 1i, 2a–d, and 4a, b, a "mitochondrial fusion factor" was calculated per sample as described above, thereby yielding a single datapoint per sample (shown as individual dots in the figures). The $t$-tests were then applied to these datapoints.

The significance of the changes seen in Fig. 1c, e, g was confirmed by using simplex plots (Supplementary Fig. 2). Simplex plots allow the depiction of data with multiple variables which sum to a constant. In this case, the fractions of neutrophils with fragmented, intermediate or fused mitochondria from one blood sample sum to 1. In the plots, each individual is color coded with a particular color, which is kept the same across timepoints (Supplementary Fig. 2). With an R package that is described in refs. [25,26], centered planar transformations were applied to map the data isometrically to a 1-dimensional Euclidian vector space, and then a linear mixed effect model was fitted for each composite to test the effect of the drinks (C18:0 vs mock). This method directly assesses the statistical significance of the difference between the C18:0 and the mock drinks in terms of changes in mitochondrial morphology across timepoints. ($n = 21$ subjects, of which 10 healthy and 11 type-2 diabetic.)

The Mann–Whitney test was used to test for significance in Supplementary Figures 6a,b,g,h and Supplementary Figure 7, since each of these figure panels represents data from a single subject. For each blood sample, 60 neutrophils were scored as follows: 1 (fragmented), 2 (intermediate), and 3 (fused). The non-parametric Mann–Whitney test was then applied in Microsoft Excel to assess the significance in the change in neutrophil score distribution between samples. Hence in these panels only, the statistical test was applied on the distribution of scores of individual neutrophils.

For Figs. 2e–i and 3c, d, correlations and their significance (two-tailed $p$) were calculated with Pearson's test in GraphPad Prism.

The significance of the lack of mitochondrial fusion caused by C16:0 was tested as follows. The null-hypothesis to be rejected was that the effect of C16:0 ingestion is non-inferior to that of C18:0 ingestion. C18:0 caused mitochondrial fusion in 19 of the 21 subjects. (Mitochondrial fusion was defined as an increase in the mitochondrial fusion factor (mff) by at least 0.19, which is 1.64 standard deviations of the mitochondrial fusion factors observed at 0 h, corresponding to a false-discovery rate of 5%.) Since 90% of all subjects (19 out of 21) responded to C18:0 ingestion, there is a 10% probability of a subject not fusing their mitochondria upon C18:0 ingestion. If C16:0 behaves like C18:0, we would expect 10% of subjects to not fuse their mitochondria by chance upon C16:0 ingestion. Instead, 5 out of 5 subjects did not fuse their mitochondria after C16:0 ingestion, which has a probability of happening of $(10\%)^5 = 10^{-5}$. Hence we reject the null hypothesis that C16:0 causes mitochondrial fusion like C18:0.

Multivariate linear regression analysis was performed with SPSS statistical analysis software with mitochondrial fusion factor as the dependent variable and all parameters shown in Supplementary Data 3 as independent variables using default parameters and confidence interval = 95%.

**Quantification of lipids with GC/MS.** Lipids were extracted from 400 µL of serum by adding 600 µL of $KH_2PO_4$ followed by 1 ml of methanol (Sigma 646377) and finally 3 ml of chloroform (Sigma 650471) in a 7 ml screw top tube (SLS TUB1202). Internal standards were added in the chloroform and were as follows: FFAs, deuterated tridecanoic acid (C13D25O2H, Cambridge Isotopes DLM1392), 20 µg; Triglycerides 1,2,3-Triheptadecanoylglycerol (Sigma), 40 µg; Phospholipids, 1,2-diundecanoyl-sn-glycero-3-phosphocholine (Avanti polar lipids), 40 µg. Samples were vortexed for 2 min then centrifuged for 10 min at 720 g. The lower layer was collected and transferred to a new 7 ml glass tube and dried under nitrogen.

Lipid classes were separated by solid phase extraction. To carry out the solid phase extraction, the lipids were resuspended in 1 ml chloroform and transferred to aminopropyl ($NH_2$) BondElut solid-phase extract columns (Agilent, 12113014). The flowthrough was collected along with two washes of 1 ml chloroform and dried under the nitrogen stream. The flowthrough contained the neutral lipid fraction. Phospholipids were eluted from the column by addition of 2 ml of 60:40 chloroform methanol. Finally FFAs were eluted using 100:2:2 Chloroform:Methanol:Glacial acetic acid. All fractions were dried under nitrogen.

To derivatize the samples into fatty acid methyl esters, 875 µL of a mixture of chloroform (Sigma 650471): methanol (Sigma 646377): boron trifluoride-methanol solution (Sigma 134821) (15:19:1 v/v/v) was added to the dried lipids in 7 ml glass tubes. The tubes were then sealed and incubated in an oven at 80 °C for 90 min. After cooling the samples, 1 ml of hexane (Sigma 34859) and 500 µL of water (Sigma 34877) were added, and the samples were vortexed and centrifuged at 720g at room temperature for 5 min. The upper organic layer was transferred into glass autosampler vials and dried under a nitrogen stream. The samples were reconstituted in 1 ml hexane prior to the gas chromatography analysis.

Gas chromatography was performed on an Agilent 7890B GC using a Thermo Scientific TR-FAME column (length: 30 m, internal diameter: 0.25 mm, film size: 0.25 µm) with helium as carrier gas (1.9 ml/min). FAMEs were detected using a 5977A MSD. Inlet and MSD temperature were set at 230 °C. The oven program temperature steps used were as follows.

The column was held at 100 °C for 2 min, then the temperature was increased to 150 °C at a rate of 25 °C/min. The temperature was further increased to 162 °C at a rate of 2.5 °C/min, with a hold time of 3.8 min. Subsequently, the temperature was increased to 173 °C at a 4.5 °C/min rate, followed by a hold time of 5 min. The column temperature was further increased to 210 °C with a 5 °C/min rate. Finally, the temperature was increased to 230 °C with a 40°C/min rate, followed by a hold time of 0.5 min. The transfer line temperature was set at 240 °C, and the ion source was at 250 °C, operating at 70 eV for electron ionization (EI). The detector was initiated after 240 s, and full scan spectra were collected over a range of 50–650 $m/z$.

To identify the FAME peaks, retention times of peaks in the samples were compared with those in external standards (Restek 35077 Food industry FAME mix and Supelco 46904 Vaccenic Methyl ester) using MS Quantitative Analysis (Agilent). The areas of specific ions for each fatty acid were used for quantification, with the background subtracted to the values of blank samples. The quantity of a given fatty acid methyl ester was determined using a standard curve generated using the external standard and normalized to the relevant internal standard for the given fraction. Data is expressed as molar%, which was calculated by dividing any individual fatty acid methyl ester by the all the detected fatty acids in a given sample.

**Quantification of acylcarnitines.** Acylcarnitines were determined in serum by electrospray ionization tandem mass spectrometry (ESI-MS/MS) according to a modified method as previously described[27], using a Quattro Ultima triple quadrupole mass spectrometer (Micromass, Manchester, UK) equipped with an electrospray ion source and a Micromass MassLynx data system. In particular, 5 µl of plasma were placed on a 4.7 mm filter paper punch, dried at room temperature overnight and extracted with 100 µl of deuterium-labeled standard solution in methanol[27].

**Other blood and serum measurements.** Methylglyoxal was measured according to ref. [28]. Human serum hepcidin was measured using the "Bioactive hepcidin 25 ELISA kit" (DRG International) following manufacturer's instructions. The hepcidin concentration was extrapolated against a standard curve by using the four parameters logistic model of the Graphpad Prism v7 software. All other blood parameters were measured by the clinical analysis facilities of the Heidelberg University Hospital Central Laboratory, which is accredited according to DIN EN ISO 15189.

**Data availability.** The authors declare that the data supporting the findings of this study are available within the paper and its supplementary information files.

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

## Acknowledgements

We thank the participants of the clinical study for their essential contribution to this work. We gratefully acknowledge help by Natalia Becker and Annette Kopp-Schneider from the DKFZ Division of Biostatistics. We thank Johannes Krisam of the Medical Biometry Department (Heidelberg University) for calculating the required case number for each group. This study was funded by an ERC Consolidator Grant "C18Signaling" (724286) to A.A.T., by a DFG SFB1118 to T.F., P.N., D.H.P., M.U.M., and A.A.T., by a DFG SFB1036 grant to M.U.M. and A.A.T., by funding from the German Center for Diabetes Research (DZD) to T.F. and P.N., and by BHF (Grant: RG/18/7/33636) and MRC grants (Grant: MRC_MC_UU_12012/2) to S.V. and A.V.-P.

## Author contributions

All the authors designed experiments. S.V. measured serum lipids, K.V.S. measured serum acyl-carnitine levels, T.F. measured serum MG levels, S.A. measured serum hepcidin levels, D.H.P. performed the clinical work including recruiting patients, maintaining records, and performing the blood sampling. D.S.-T. performed all other experiments including mitochondrial morphology measurements, sample management, and statistical analyses. All the authors analyzed data. D.S.-T., D.H.P., S.V., T.F., M.U.M., J.G.O., A.V.-P., P.N., and A.A.T. wrote the manuscript.

## Additional information

**Competing interests:** The authors declare no competing interests.

