## [Peer Review File · Nature Communications]

Reviewers' comments:

Reviewer #2 (Remarks to the Author):

This paper presents in vivo findings in humans that confirm effects of stearic acid (C18:0 fatty acid) on mitochondrial structure and function that the authors previously demonstrated in drosophila and cellular models. The data are compelling and are supported by their earlier work documenting the role of stearylolation of transferrin receptor 1 in triggering the mitochondrial effects.

While the use of a dietary challenge in this report shows remarkable acute effects of stearic acid on mitochondrial properties, this study design does not allow any inference to be drawn regarding the effects of longer-term consumption. In fact the failure to observe differences in C18:0 plasma levels between vegetarians and omnivores, while based on small numbers, suggests that short term changes after a stearic acid bolus may not reflect effects of chronic intake. In this regard, the heavy emphasis placed by the authors on the relevance of the present findings for dietary effects on chronic disease risk is inappropriate. A related issue is the difficulty of determining to what extent the effects of C18:0 intake can be dissociated from those of other constituents in the foods in which this fatty acid is consumed (including other fatty acids not tested here).

Another concern is the very small study population. While sufficient to demonstrate significant effects of C18:0 intake, there may be inadequate power to rule out smaller effects of the C16:0 control. In this regard, the statistical analyses do not include power estimates or formal comparisons of the results with the two fatty acids.

Specific comments:

p. 6, middle and p. 17 top: Since the classification of mitochondria into the 3 categories appears to be largely subjective, more information should be provided regarding how this was carried out – specifically, were the readings confirmed by more than one of the investigators?

p. 7, middle: On what basis was the “mitochondrial fusion factor” developed? It would be reassuring to know that this was not designed as a means of amplifying statistical significance of the findings.

p. 7, 2nd last sentence: It seem that only human HeLa cells were tested in reference 16, so the extrapolation of the present results from neutrophils to other cell types may not be justified.

p. 10, line 14: technically the diet was also low in nutrients other than C18:0.

Page 12, “We had four vegetarians in our study (ED Fig 1a)”: This info was not found in the figure.

p. 12, bottom: It is not clear how the present results shed light on the relation of C16:0 intake to cancer and cardiovascular risk.

p. 14, bottom and 15, top: Was the washout diet the same as the vegan baseline diet? If not, there could be a diet order effect. In any case, a possible diet order effect should be tested.

Page 16, second paragraph: “tubes” is repeated.

ED Figure 1a: “11 completed the first arm of cross-over” should be “11 completed both arms of cross-over”

Figure 2: An explanation should be provided for the normalization of C18:0-TAG to C16:0-TAG.

ED Figure 2: The dot sizes should be increased to make them more visible.

ED Figure 5: Might the authors comment on the trend to increased hepcidin levels after the vegan diet phase?

ED Tables 1 and 2: These were not provided.

Reviewer #3 (Remarks to the Author):

This is a very nicely designed and executed body of work. Just have a few minor points:

1) blood-drawing is not "non-invasive". It's just minimally invasive.

2) the sentence "Although clinical trials often require hundreds or thousands of people to prove a biological effect, we obtained statistically significant results with only 10 healthy subjects." is a little gratuitous. Clinical trials are more complicated for additional reasons (looking for side-effects, etc).

3) There was a small, but consistent increase in mitochondrial fusion with the mock drink. Can the authors provide some information on how much C18 is in "low-fat milk" and a small banana? Is it some? None? How much? Why didn't the authors use fat-free milk and perhaps some artificial banana flavoring?

4) Insulin stimulation also causes mitochondrial fusion (in fibroblast cell lines). Is there an data on whether fasted individuals who then ingest glucose see similar changes in fusion? Is this a transient physiological process that is being monitored, rather than a prolonged one that would have significant physiological effects?

5) Not all cells undergo routine fission and fusion. It might be an overstatement to project that the findings seen here with neutrophils would apply to "most" tissues. Compounding this, whether the changing serum levels of C18:0 would result in similar changes in c18:0 in interstitial fluids would need to be confirmed.

Reviewer #4 (Remarks to the Author):

The manuscript of Senyilmaz-Tiebe et al describes a small clinical trial to evaluate the effect stearic acid on mitochondrial structure and function. It builds on earlier work examining the effect of stearic acid on mitochondrial function in cell culture and in drosophila models. While this is an interesting and exciting effect identified by the authors, I think that the study described here attempting to relate this to human diet has a number of problems and limitations.

Overall the manuscript is well written and the introduction quite appropriate and informative.

The study design itself has some limitations as the control arm is (no fat) and so the interpretation does not allow the effect shown to be assigned to stearic acid, but could be a general effect of any fatty acid (in the human diet setting). Ideally a palmitic acid arm would have been completed on the same individuals either as well as or instead of the no fat arm.

The authors attempt to address the absence of a palmitate control by the additional testing of a single participant with palmitate. While the results of this appear to support the claims made

around stearic and palmitic acid, in this analysis, the 0 time point of the C 16:0 treatment is quite different from the 0 time point of the C 18:0 treatment placing doubt on any interpretation. However, more importantly, with an n=1 it is not appropriate to report this result. It becomes misleading as the authors place weight on the outcome when there is no statistical significance at all. This cannot be used to support any other findings.

The correlation analysis shown in Fig 2D-F and 3B are also misleading as the analysis is using repeat measures of 20 individuals to increase power. These are not independent events and so should not be treated as such. If the goal is to check the association between the mito fusion factor and C18:0 TAG then this should be done on a cohort large enough to provide a significant finding. If the goal is to show that the change in mito fusion correlated with the change in C18:0 TAG then an appropriate analysis should be performed. Perhaps an analysis of the deltas for each. In any event the correlation analysis as shown is not appropriate.

This also highlights the very limited power of this study, which while interesting is underpowered and suffers from some serious design flaws.

Reviewer #2:

This paper presents in vivo findings in humans that confirm effects of stearic acid (C18:0 fatty acid) on mitochondrial structure and function that the authors previously demonstrated in drosophila and cellular models. The data are compelling and are supported by their earlier work documenting the role of stearylolation of transferrin receptor 1 in triggering the mitochondrial effects.

While the use of a dietary challenge in this report shows remarkable acute effects of stearic acid on mitochondrial properties, this study design does not allow any inference to be drawn regarding the effects of longer-term consumption. In fact the failure to observe differences in C18:0 plasma levels between vegetarians and omnivores, while based on small numbers, suggests that short term changes after a stearic acid bolus may not reflect effects of chronic intake. In this regard, the heavy emphasis placed by the authors on the relevance of the present findings for dietary effects on chronic disease risk is inappropriate.

We have now toned down the text in the Discussion on the possible relevance of long-term dietary effects on chronic disease. That said, we do believe it is important to cover the relevant literature linking C18:0 to human physiology and pathophysiology in the Introduction, so we have not removed citations from the Introduction. In fact Reviewer 4 writes that our Introduction is nice and covers the topic well.

That said, there are several important points to consider before concluding that these results are not relevant for chronic disease risk:

1. All our blood samples were collected after an overnight starvation, and no breakfast. Hence this is a baseline amount of C18:0 in people in the absence of recent food ingestion. As soon as people eat, the data in Figure 2 show that circulating C18:0 levels increase in response to dietary intake of C18:0, and this will be different in vegetarians versus omnivores. Hence it is most likely that if people were sampled randomly during the day, on average vegetarians would have lower C18:0 levels than omnivores. We have now made this more clear in the text.

2. Although diet may or may not have long-term effects on circulating C18:0 levels, nonetheless genetic differences between people could very well affect circulating C18:0 levels, and this could then impact chronic disease risk.

A related issue is the difficulty of determining to what extent the effects of C18:0 intake can be dissociated from those of other constituents in the foods in which this fatty acid is consumed (including other fatty acids not tested here).

If we understand correctly, the reviewer is pointing out that the epidemiological studies on C18:0 have this issue. (Because regarding our study, in ED Fig. 6a we see mitochondrial fusion with pure C18:0, without the banana or the milk, so this excludes interactions between the C18:0 and other components of the drink). We agree. It is exactly for this reason that studies such as ours are needed, because they are very controlled and test specifically for response to C18:0 (since our negative control is mock drink specifically lacking C18:0).

Another concern is the very small study population. While sufficient to demonstrate significant effects of C18:0 intake, there may be inadequate power to rule out smaller effects of the C16:0 control. In this regard, the statistical analyses do not include power estimates or formal comparisons of the results with the two fatty acids.

We thank the reviewer for this comment, prodding us to look at C16:0 in more depth. To address this, we have now performed an additional arm to the study. After consulting with our biometrician, we applied for permission to ask the previous healthy participants of our study whether they wanted to participate in another study arm where they would receive either C16:0, C18:0 or mock. We selected the 0h and 3h hour post-ingestion timepoints to analyze mitochondrial morphology, because the 3h timepoint showed the best response. After receiving permission from our ethics committee, we asked the previous participants and were able to re-recruit 6 healthy controls. Four people were given C16:0 drinks, one person was given a C18:0 drink (as a positive control) and one person was given a mock drink lacking C18:0 or C16:0 (as a negative control). Importantly, we did this in a double-blind fashion, and the person scoring mitochondrial morphology did not know how many people received which drink, nor which person received which drink. These new results, presented in ED Fig. 6c-c'' are very clear. The positive-control and negative-control people responded as expected (ED Figs. 6c and 6c'', respectively). All four people ingesting the C16:0 drink did not show mitochondrial fusion (ED Fig. 6c), so that in total all 5 people ingesting C16:0 did not show mitochondrial fusion. Since only 10% of people in the main study did not respond to C18:0, this negative result for C16:0 ingestion is highly significant ($p=10^{-5}$).

Specific comments:

p. 6, middle and p. 17 top: Since the classification of mitochondria into the 3 categories appears to be largely subjective, more information should be provided regarding how this was carried out – specifically, were the readings confirmed by more than one of the investigators?

We have added a more detailed description to the Materials & Methods. Cells containing a single large interconnected network were scored as ‘fused’, cells containing individual dots were score as ‘fragmented’, and all other cells were scored as ‘intermediate’. We were forced to perform the study in this manner because we first tried various protocols for fixing the blood, but these protocols caused the mitochondrial morphology to be unclear. (The images were ‘fuzzy’). Fixation would have allowed more time-consuming imaging of the mitochondria which we could have used to bioinformatically quantify fragmentation, as we have done in the past (PMID 26214738). Without fixation, however, we needed to score morphology quickly, and the method we used allowed us to score morphology within 30 minutes of blood collection. We have added a comment to this effect in the Materials & Methods.

One investigator scored all mitochondrial morphologies, in order to have consistent scoring across all samples. Importantly, this was done in a blinded fashion. In addition, we now asked a second person to also score some of the new samples presented in ED Fig. 6c-c”, and the results are similar (new ED Fig. 7). This has now been added to the Materials & Methods.

p. 7, middle: On what basis was the “mitochondrial fusion factor” developed? It would be reassuring to know that this was not designed as a means of amplifying statistical significance of the findings.

The ‘mitochondrial fusion factor’ does not amplify the significance of the findings – the results are highly statistically significant also if we statistically analyze the changes in morphology profiles show in Fig 1c directly ($p \leq 0.0001$), or using a simplex plot analysis (ED Fig. 2, $p \leq 0.001$ or $p < 0.0001$ depending on which parameter). The mitochondrial fusion factor is defined using a very intuitive definition: all cells that have a ‘fused’ morphology get counted as ‘fused’. All intermediate cells get counted in the ‘mitochondrial fusion factor’ as being half as fused (ie weighting factor of 1, compared to a weighting factor of 2 for the fully fused cells).

p. 7, 2nd last sentence: It seem that only human HeLa cells were tested in reference 16, so the extrapolation of the present results from neutrophils to other cell types may not be justified.

We have also tested HEK293 cells, as well as Drosophila S2 cells, and all these cells respond the same way. Nonetheless, we have softened the wording of this sentence by changing it from “it is likely” to “it is possible”.

p. 10, line 14: technically the diet was also low in nutrients other than C18:0.

We have fixed it to “low-fat vegan diet”.

Page 12, “We had four vegetarians in our study (ED Fig 1a)”: This info was not found in the figure.

We have added the information to ED Fig 1b, and fixed the citation.

p. 12, bottom: It is not clear how the present results shed light on the relation of C16:0 intake to cancer and cardiovascular risk.

We have now made this more explicit:

“If dietary C18:0 signals the intake of lipids to the human body, to activate a physiological response for lipid handling which includes fatty acid beta-oxidation, whereas C16:0 does not, this would imply that C16:0 ingestion will lead to more fat accumulation in the body than C18:0 ingestion. Fat accumulation, in turn, is a risk factor for both cardiovascular disease and cancer.”

p. 14, bottom and 15, top: Was the washout diet the same as the vegan baseline diet? If not, there could be a diet order effect. In any case, a possible diet order effect should be tested.

The wash-out phase consisted of the normal diet of the respective person, and it was followed by 2 days of low-fat vegan diet, just like for the first arm of the study. (If the washout diet had been the vegan diet, this would have introduced an effect because the subjects would have been on a vegan diet for >5 days, unlike the first arm of the study). Furthermore, we know from Rhee et al (1997) that ingested C18:0 is cleared from circulation on the order of hours after ingestion, so that we would not expect a diet-order effect with a wash-out phase of at least 3 days.

To look at a possible diet order effect on the response to either the C18:0 drink or the mock drink, we calculated separately the change in mitochondrial morphology at 3h vs 0h for the subjects who either received the C18:0 drink

first, or those who received the mock drink first (Reviewer Figure 1 below). C18:0 ingestion caused a ~2-fold increase in mitochondrial fusion in both groups of subjects, and the difference was not statistically significant. Likewise, the response to the mock drink was also not significantly different in the two groups.

Page 16, second paragraph: "tubes" is repeated.

Fixed – thanks !

ED Figure 1a: "11 completed the first arm of cross-over" should be "11 completed both arms of cross-over"

Fixed – thanks !

Figure 2: An explanation should be provided for the normalization of C18:0-TAG to C16:0-TAG.

We have added to the figure legend that we normalized to C16:0 because it is both abundant (ie good signal-to-noise) and stable.

ED Figure 2: The dot sizes should be increased to make them more visible.

Fixed – thanks !

ED Figure 5: Might the authors comment on the trend to increased hepcidin levels after the vegan diet phase?

The reviewer makes an interesting observation, but we don't know why that would be the case. We have added a sentence to the Results section pointing this out.

ED Tables 1 and 2: These were not provided.

We have uploaded these files to the online submission system.

Reviewer #3:

This is a very nicely designed and executed body of work. Just have a few minor points:

We thank the reviewer for this positive evaluation !

1) blood-drawing is not "non-invasive". It's just minimally invasive.

Fixed – thanks !

2) the sentence "Although clinical trials often require hundreds or thousands of people to prove a biological effect, we obtained statistically significant results with only 10 healthy subjects." is a little gratuitous. Clinical trials are more complicated for additional reasons (looking for side-effects, etc).

We did not mean to be disparaging, but wanted to point out that the effect is quite robust. We have removed this statement.

3) There was a small, but consistent increase in mitochondrial fusion with the mock drink. Can the authors provide some information on how much C18 is in "low-fat milk" and a small banana? Is it some? None? How much? Why didn't the authors use fat-free milk and perhaps some artificial banana flavoring?

Indeed the mock drink has roughly 0.5 grams of C18:0, which could explain the small increase in mitochondrial fusion. We have added this to the text. (The purpose of the banana was not only for flavor, but also to help emulsify the fat.)

4) Insulin stimulation also causes mitochondrial fusion (in fibroblast cell lines). Is there an data on whether fasted individuals who then ingest glucose see similar changes in fusion? Is this a transient physiological process that is being monitored, rather than a prolonged one that would have significant physiological effects?

We are not aware of any data regarding mitochondrial fusion in response to glucose ingestion. In our hands, we do not see a correlation between insulin levels and mitochondrial fusion in neutrophils (ED Table 2). Another way to see this is that in our samples insulin increases on average at the 6h timepoint, likely due to the low-fat lunch that was eaten after the 3h timepoint. However, between 3h and 6h, there is no increase in fusion – if anything, there is a drop. Hence we

think the mild increase in mitochondrial fusion in response to the mock drink is more likely due to the 0.5g C18:0 contained in the mock drink, as suggested by the reviewer in the previous point.

5) Not all cells undergo routine fission and fusion. It might be an overstatement to project that the findings seen here with neutrophils would apply to "most" tissues. Compounding this, whether the changing serum levels of C18:0 would result in similar changes in c18:0 in interstitial fluids would need to be confirmed.

We agree and have toned down the statement, so that it now reads

“Hence we believe it is possible that other cells in the human body may also fuse their mitochondria after a meal containing C18:0, although future work will be required to test this.”

Reviewer #4:

The manuscript of Senyilmaz-Tiebe et al describes a small clinical trial to evaluate the effect stearic acid on mitochondrial structure and function. It builds on earlier work examining the effect of stearic acid on mitochondrial function in cell culture and in drosophila models. While this is an interesting and exciting effect identified by the authors, I think that the study described here attempting to relate this to human diet has a number of problems and limitations. Overall the manuscript is well written and the introduction quite appropriate and informative.

We thank the reviewer for the positive assessment regarding the writing and the introduction.

The study design itself has some limitations as the control arm is (no fat) and so the interpretation does not allow the effect shown to be assigned to stearic acid, but could be a general effect of any fatty acid (in the human diet setting). Ideally a palmitic acid arm would have been completed on the same individuals either as well as or instead of the no fat arm. The authors attempt to address the absence of a palmitate control by the additional testing of a single participant with palmitate. While the results of this appear to support the claims made around stearic and palmitic acid, in this analysis, the 0 time point of the C 16:0 treatment is quite different from the 0 time point of the C 18:0 treatment placing doubt on any interpretation. However, more importantly, with an n=1 it is not appropriate to report this result. It becomes misleading as the authors place weight on the outcome when there is no statistical significance at all. This cannot be used to support any other findings.

Firstly, we would like to point out that we can indeed assign the effect to stearic acid, because this is the only variable that changes between the intervention and the control arm. Hence it must be due to the stearic acid.

However, the reviewer raises a separate point, which is whether this effect is specific to C18:0, or whether it would also be seen with C16:0. To address this, we performed what the reviewer suggests. After consulting with our biometrician, we applied for permission to ask the previous healthy participants of our study whether they wanted to participate in another study arm where they would receive either C16:0, C18:0 or mock. We selected the 0h and 3h hour post-ingestion timepoints to analyze mitochondrial morphology, because the 3h timepoint showed the best response. After receiving permission from our ethics committee, we asked the previous participants and were able to re-recruit 6 healthy controls. Four people were given C16:0 drinks, one person was given a C18:0 drink (as a positive control) and one person was given a mock drink

lacking C18:0 or C16:0 (as a negative control). Importantly, we did this in a double-blind fashion, and the person scoring mitochondrial morphology did not know how many people received which drink, nor which person received which drink. These new results, presented in ED Fig. 6c-c'' are very clear. The positive-control and negative-control people responded as expected (ED Figs. 6c and 6c'', respectively). All four people ingesting the C16:0 drink did not show mitochondrial fusion (ED Fig. 6c), so that in total all 5 people ingesting C16:0 did not show mitochondrial fusion. Since only 10% of people in the main study did not respond to C18:0, this negative result for C16:0 ingestion is highly significant ($p=10^{-5}$).

The correlation analysis shown in Fig 2D-F and 3B are also misleading as the analysis is using repeat measures of 20 individuals to increase power. These are not independent events and so should not be treated as such. If the goal is to check the association between the mito fusion factor and C18:0 TAG then this should be done on a cohort large enough to provide a significant finding. If the goal is to show that the change in mito fusion correlated with the change in C18:0 TAG then an appropriate analysis should be performed. Perhaps an analysis of the deltas for each. In any event the correlation analysis as shown is not appropriate.

Firstly, we would like to point out that the effect is highly significant also in Fig 1c, 1d, 1e and ED Fig. 2 where the 3h or the 6h timepoints are compared individually to the 0h timepoint.

We respectfully disagree conceptually, however, with the reviewer's objection. In theory, one could imagine doing an entire study on one single person, taking blood every hour over an entire week, and showing a correlation between circulating C18:0 levels and mitochondrial fusion. This would be a valid approach, and would rely entirely on sampling the same individual at multiple times.

Perhaps important to note is that we are not including repeat measurements on the same sample, and that the 6h timepoint is quite different from the 3h timepoint. (It is as different to the 3h timepoint as the 3h timepoint is from the 0h timepoint, both time-wise and if one looks at the circulating C18:0 levels in Fig. 2b).

As suggested by the reviewer, we have also performed the delta analysis, and this is also significant (Reviewer Figure 2 below):

Reviewer Figure 2: The increase in mitochondrial fusion factor from 0h to 3h correlates with the increase in serum C18:0 levels from 0h to 3h, both if we consider all datapoints (upper panel) and if we exclude the three best responding points (lower panel).

This also highlights the very limited power of this study, which while interesting is underpowered and suffers from some serious design flaws.

We are a bit perplexed by this comment, because by definition, the study cannot be underpowered if it results in statistically significant findings (e.g. Fig 1c).

Indeed, the study size was designed in consultation with a biometrician, Johannes Krisam (in the Acknowledgements), based on data from a pilot study, and using formulas from Welleck & Blettner 2012 for cross-over studies. These calculations showed that for reaching significance of 0.01 and a power of 95% 10 participants per group are required. For this reason we recruited 20 subjects per group (giving oral consent) assuming there would be roughly a 50% drop-out rate.

Reviewers' comments:

Reviewer #2 (Remarks to the Author):

The authors have satisfactorily addressed this reviewer's concerns, There is only one very minor point, arising from the fact that I had not received ED Table 2 with the original ms. There are nominally significant correlations of change in mitochondrial fusion with erythrocytes and basophil granulocytes, and this is not acknowledged in the text (p. 9, lines 9-11). One way to deal with this is to say that "there were no significant correlations with any of the measured parameters after accounting for multiple testing".

Reviewer #3 (Remarks to the Author):

The authors adequately addressed my comments.

Reviewer #4 (Remarks to the Author):

The Authors have addressed my concerns by performing additional experiments to test the effect of palmitic acid in mito fusion. This is a good start and it certainly looks like there is no effect from palmitate. However, I am still unsure of how these data were analysed to reach a statistically significant effect with only a n=5 (or 6 if the no FA meal is included).

The authors state "Since only 10% of people in the main study did not respond to C18:0, this negative result for C16:0 ingestion is highly significant ($p=10^{-5}$)."

In the legend to Fig ED6 the authors state "Mitochondrial morphology does not change in 4 subjects who received a banana milk shake containing an equimolar amount of C16:0 corresponding to the 24g of C18:0 used in the C18:0 milkshake (n=60 per sample, n.s. $p>0.05$ of student's t test). (c') Neutrophil mitochondria fuse in one 'positive control' subject in response to C18:0 ingestion (**** $p<0.0001$ by Mann-Whitney test). (c'') Mitochondria do not fuse in one 'negative control' subject receiving a mock drink without added C16:0 or C18:0 (n.s. $p>0.05$ by Mann-Whitney test).

I take it from this, that the authors are performing the statistics not on the individual who performed the test but on a number (60?) of measurements that are made on each individual. Then to determine if the one individual who had the steric acid was different from the five individuals who did not they compared the 60 measurements of the one against the 300 measurements of the five in a Mann Whitney U test. Is that correct? If so I think this is also inappropriate resampling of the data to reach significance. One of my original problems with the analysis. Alternately the authors may have compared the four palmitate treated individuals against the initial stearate treated individuals, if this was the case this should be clearly stated.

This also appears to have been done in the Reviewer Figure 2 where the increase in Mito fusion factor from 0-3 hours is plotted against the delta C18:0 TAG. There are ~80 points plotted in this analysis to give a $p<0.01$. However there were only 20 participants who completed both arms of the study, so that gives 40 data points that could be plotted here. Perhaps I am missing something, where do the 80 points come from and how were they calculated?

This comes back to my initial point of using repeated measures in a correlation analysis, The authors state that "In theory one could imagine doing an entire study on one single person, taking blood every hour over an entire week, and showing a correlation between circulating C18:0 levels and mitochondrial fusion" While this would be possible, but would only address the question of the

correlation in that single individual. I was assuming that the authors were trying to demonstrate an effect in the test population.

Perhaps it would help if the authors were to provide a statistical analysis section in the methods where the respective analyses are clearly described, including sampling of the data for each analysis.

Reviewer #2:

The authors have satisfactorily addressed this reviewer's concerns, There is only one very minor point, arising from the fact that I had not received ED Table 2 with the original ms. There are nominally significant correlations of change in mitochondrial fusion with erythrocytes and basophil granulocytes, and this is not acknowledged in the text (p. 9, lines 9-11). One way to deal with this is to say that "there were no significant correlations with any of the measured parameters after accounting for multiple testing".

We thank the reviewer for the positive assessment. We have changed the sentence as suggested by the reviewer.

Reviewer #3:

The authors adequately addressed my comments.

We thank the reviewer for the positive assessment.

Reviewer #4:

The Authors have addressed my concerns by performing additional experiments to test the effect of palmitic acid in mito fusion. This is a good start and it certainly looks like there is no effect from palmitate. However, I am still unsure of how these data were analysed to reach a statistically significant effect with only a n=5 (or 6 if the no FA meal is included).

The authors state "Since only 10% of people in the main study did not respond to C18:0, this negative result for C16:0 ingestion is highly significant ($p=10^{-5}$)."

In the legend to Fig ED6 the authors state "Mitochondrial morphology does not change in 4 subjects who received a banana milk shake containing an equimolar amount of C16:0 corresponding to the 24g of C18:0 used in the C18:0 milkshake (n=60 per sample, n.s. $p>0.05$ of student's t test). (c') Neutrophil mitochondria fuse in one 'positive control' subject in response to C18:0 ingestion (**** $p<0.0001$ by Mann-Whitney test). (c'') Mitochondria do not fuse in one 'negative control' subject receiving a mock drink without added C16:0 or C18:0 (n.s. $p>0.05$ by Mann-Whitney test).

I take it from this, that the authors are performing the statistics not on the individual who performed the test but on a number (60?) of measurements that are made on each individual. Then to determine if the one individual who had the steric acid

was different from the five individuals who did not they compared the 60 measurements of the one against the 300 measurements of the five in a Mann Whitney U test. Is that correct? If so I think this is also inappropriate resampling of the data to reach significance.

No, this is not how we calculated the significance; we apologize for the lack of clarity. We have now included a new section in the Materials & Methods that explains how the statistical tests were performed.

There are two separate significance questions related to ED Fig 6c. The first is whether the distribution in mitochondrial morphologies seen at timepoint 3h is different from that seen at 0h (ie whether C16:0 ingestion causes mitochondrial fusion). The second is whether the effect of C16:0 (seen in ED Fig 6c) is significantly different from the effect of C18:0 (seen in main Fig 1c).

1) does C16:0 cause fusion?

To address this, for each blood sample (5 subjects x 2 timepoints = 10 samples) we scored the mitochondrial morphology of 60 neutrophils as either 'fragmented', 'intermediate' or 'fused'. From this we calculated for each sample the percentage of neutrophils in each of the three categories. These percentages were then combined to generate the graph shown in ED Fig 6c (e.g. at 0h, the 'fragmented' bar shows the average and standard deviation of 5 datapoints, representing the % of fragmented neutrophils in the 5 subjects at 0h). Analogously, this was done in Fig 1c for C18:0, where the first bar on the left of the graph shows the average and standard deviation of 21 datapoints, representing the % of fragmented neutrophils in the 21 subjects that participated in the study. To compare the 3h to the 0h timepoints in ED Fig 6c, we performed a 2-tailed paired student t-test on the 5 'fragmented' datapoints at time 0h versus the 5 'fragmented' datapoints at time 3h. One can see also by eye that this is not significant. In comparison, in main Fig 1c, a t-test on the 21 'fragmented' datapoints of 3h compared to the 21 'fragmented' datapoints at 0h gives $p < 0.0001$. The reviewer will note that this test only compares the fraction of cells that are 'fragmented' in the two timepoints. However, a similar comparison of the 'intermediate' or of the 'fused' fractions in ED Fig 6c would yield the same result.

2) is the effect of C16:0 different from the effect of 18:0?

To address this, we calculated the mitochondrial fusion factor (mff) for each blood sample, which simplifies the situation by combining the %fragmented, %intermediate and %fused (ie 3 values) into one single value for each blood sample. In addition, the mff is sensitive to any change between these three categories (whereas performing t-tests on only one category, such as %fragmented, ignores shifts between the other two categories). We see that C18:0 causes mitochondrial fusion in 19 out of the 21 subjects, with 'fusion' defined as an increase in the mff by at least 0.19 (which is 1.64 standard deviations of the mitochondrial fusion factors observed at 0h, corresponding to a false-discovery rate of 5%). Since 90% of all subjects (19 out of 21) responded to

C18:0 ingestion, there is a 10% probability of a subject not fusing their mitochondria upon C18:0 ingestion. We then postulate the null hypothesis to be rejected, which is that the effect of C16:0 ingestion is non-inferior to that of C18:0 ingestion. If C16:0 behaves like C18:0, we would expect 10% of subjects to not fuse their mitochondria by chance upon C16:0 ingestion. Instead, 5 out of 5 subjects did not fuse their mitochondria after C16:0 ingestion (ie all 5 subjects did not increase their mff by ≥ 0.19 .) This has a probability of happening of $(10\%)^5 = 10^{-5}$. Hence we reject the null hypothesis that C16:0 causes mitochondrial fusion like C18:0. (ie the statistics was indeed done on the 5 individuals, and there is no data resampling).

One of my original problems with the analysis. Alternately the authors may have compared the four palmitate treated individuals against the initial stearate treated individuals, if this was the case this should be clearly stated.

Yes. (5 palmitate treated individuals in total). Please see above and an explanation has been added to the "Statistical Analyses" section of the Materials & Methods.

This also appears to have been done in the Reviewer Figure 2 where the increase in Mito fusion factor from 0-3 hours is plotted against the delta C18:0 TAG. There are ~80 points plotted in this analysis to give a $p < 0.01$. However there were only 20 participants who completed both arms of the study, so that gives 40 data points that could be plotted here. Perhaps I am missing something, where do the 80 points come from and how were they calculated?

We thank the reviewer for catching this. Indeed, we made a mistake in Reviewer Figure 2 and included both the (3h - 0h) and the (6h-0h) datapoints, which is why there are 84 datapoints instead of 42 (21 participants x 2 arms of the study). The corrected Reviewer Figure is provided below, e.g. with only the (6h-0h) data:

This comes back to my initial point of using repeated measures in a correlation analysis, The authors state that "In theory one could imagine doing an entire study on one single person, taking blood every hour over an entire week, and showing a correlation between circulating C18:0 levels and mitochondrial fusion" While this would be possible, but would only address the question of the correlation in that single individual. I was assuming that the authors were trying to demonstrate an effect in the test population.

Perhaps it would help if the authors were to provide a statistical analysis section in the methods where the respective analyses are clearly described, including sampling of the data for each analysis.

As suggested, we have now included a section entitled "Statistical Analyses" to the Materials & Methods. We hope this clarifies the statistical methods that were used.

REVIEWERS' COMMENTS:

Reviewer #4 (Remarks to the Author):

The authors have addressed my concerns. A nice paper.